# Oral Health Status of Adult Dysphagic Patients That Undergo Endoscopic Gastrostomy for Long Term Enteral Feeding

**DOI:** 10.3390/ijerph19084827

**Published:** 2022-04-15

**Authors:** Sara Lopes, Vitor Tavares, Paulo Mascarenhas, Marta Lopes, Carolina Cardote, Catarina Godinho, Cátia Oliveira, Carla Adriana Santos, Madalena Oom, José Grillo-Evangelista, Jorge Fonseca

**Affiliations:** 1Centro de Investigação Interdisciplinar Egas Moniz, Instituto Universitário Egas Moniz (IUEM), 2829-511 Almada, Portugal; saradeoliveiralopes@outlook.com (S.L.); vitotav@hotmail.com (V.T.); marta.oliveiralopes@hotmail.com (M.L.); carolinacardoteduarte@gmail.com (C.C.); moom@egasmoniz.edu.pt (M.O.); josegrillo87@gmail.com (J.G.-E.); 2Grupo de Patologia Médica, Nutrição e Exercício Clínico (PaMNEC) do Centro de Investigação Interdisciplinar Egas Moniz (CiiEM), 2829-511 Almada, Portugal; pmascarenhas@egasmoniz.edu.pt (P.M.); cgcgodinho@gmail.com (C.G.); 3Artificial Feeding Team (GENE), Gastroenterology Department, Hospital Garcia de Orta, 2805-267 Almada, Portugal; sofi.doliveira@gmail.com (C.O.); carla.adriana.santos@hotmail.com (C.A.S.)

**Keywords:** dysphagia, enteral feeding, percutaneous endoscopic gastrostomy, oral health

## Abstract

Background: Endoscopic Gastrostomy (PEG) is required to maintain a feeding route when neurological or cancer dysphagia impact oral intake. This study aimed to evaluate the oral health and oral changes of PEG-patients without oral feeding for three months. Methods: Prospective observational study, with a PEG-patients convenience sample. Data were obtained before PEG (T0) and 3 months after gastrostomy (T1). Initial oral hygiene habits were collected through a questionnaire. Intra-oral evaluation was performed using: Plaque Index (IP), Gingival Index (IG), Decayed, Missing and Filled Teeth Index (DMF), Community Periodontal Index (CPI), and Attachment Loss (AL). T0 and T1 were compared to evaluate oral health evolution. Results: Thirty-nine patients aged 65.3 ± 17.4 years were included. Initial (T0) oral health was worse than expectable. Between assessments period, oral indexes suffered a general deterioration with statistical relevance to the DMF. The frequency of deep periodontal pockets and attachment loss remained stable. Conclusions: PEG-patients presented poor oral health and insufficient oral hygiene habits, even before gastrostomy. After three months of PEG feeding, oral health suffered a general deterioration. This outcome was probably associated with the absence of oral feeding activity, which is beneficial to oral homeostasis, and further reduced oral hygiene. Improved oral daily care and dental appointments should become part of the PEG-patients follow-up.

## 1. Introduction

Dysphagia may occur as a consequence of an obstructive disease or in the setting of a neurological disorder. It reduces oral intake by decreasing swallowing efficacy and safety, leading to depletion of nutrient intake and, consequently, to malnutrition [1]. If oral intake is insufficient or the patient cannot eat or drink safely, and there is no other disorder of the digestive tract, tube feeding is the obvious feeding option [2]. Percutaneous endoscopic gastrostomy (PEG) is a simple and safe method of providing enteral access for patients with dysphagia if tube feeding is required for a long term, i.e., for longer than 3–4 weeks [3,4,5]. Although dysphagic PEG-patients present a wide range of underlying disorders, some of these features are common; they tend to be older patients, with severe disorders leading to dysphagia and several comorbidities. Additionally, they tend to be fragile patients, with other long-standing disorders, causing malnutrition long before the gastrostomy procedure. Tobacco smoking and excessive alcohol consumption are common in head and neck cancer patients [6] as well as other neurologic illnesses that lead to dysphagia, and are sometimes linked to social disruption.

All of these features may contribute to poor oral care and impaired oral health long before the beginning of enteral feeding.

Due to the individual’s frailty and short lifespan, medium and long-term longitudinal studies with PEG-patients are scarce, and studies on these patient’s oral health are even less frequent. In fact, people under long term enteral feeding through endoscopic gastrostomy are more commonly evaluated only once in cross-sectional studies. These are patients with severe diseases and very poor health conditions.

Caregivers and health professionals often overlook all health issues including the major disorders that lead to enteral nutrition. The scarcity of published studies about the oral health of adult PEG-patients may be explained by a couple of factors: patients have very serious disorders, and the treatment teams for these individuals concentrate on enteral nutrition and these primary abnormalities, ignoring other health issues.

Since a considerable number of adult PEG patients have advanced conditions and a shorter life expectancy, dental health may appear less important. Notably, many PEG patients have poor self-care capacities, and patients, caregivers, and even professional teams caring for long-term enteral feeding patients are usually unaware of the importance of dental health as a major component in overall health, preventing low-grade oral inflammation that is associated with poor outcome in a wide range of systemic disorders.

Nevertheless, an important proportion of these patients survive for years and some of them improve or heal entirely from the underlying condition, resuming oral feeding. As such, paying special attention to general health issues is critical.

Various aspects are typically disregarded, such as electrolytes [7], trace elements [8,9,10,11,12], and vitamins [13,14].

This study aims to contribute to understanding oral health in PEG-subjects, to raise awareness of the importance of this issue, and to identify strategies to minimize the degradation of oral health in patients that undergo PEG-feeding.

## 2. Materials and Methods

This is a prospective observational study, focused on patients undergoing prolonged enteral feeding by percutaneous endoscopic gastrostomy (PEG).

### 2.1. Sampling and Recruitment

The study used a convenience sample of adult patients undergoing enteral feeding through PEG recruited from the Outpatient Clinic of Artificial Nutrition performed by the Artificial Feeding Team (GENE) of the Hospital Garcia de Orta (HGO).

The inclusion criteria consisted of adult patients of both sexes with an underlying disorder such as Neurological Dysphagia or Dysphagia due to Head and Neck Cancer who had a percutaneous endoscopic gastrostomy (PEG) for prolonged enteral feeding and who had eaten orally for the month before to the gastrostomy procedure.

The exclusion criteria included: (1) refusal to participate in the study; (2) edentulous patients; (3) patients without any of the teeth usually numbered as “16”, “21”, “24”, “36”, “41”, “44”; (4) patients who maintained oral food intake after the gastrostomy, although insufficient; (5) and patients for whom it was clear that the chosen method would be impossible to implement.

Regarding recruitment and sampling, patients included in this study were all that were available in this center meeting the inclusion criteria during the time interval in which the study took place.

### 2.2. Ethical Considerations

This study followed the principles of the Declaration of Helsinki. Before conducting this study, a research protocol was approved by the Ethics Committee of the Hospital Garcia de Orta identified with the number 51/2014. Patients and/or their legal representatives were previously informed of the nature of the study and all signed written consent agreeing to participate. All of the data were collected anonymously, without intervention of the researcher and free funding.

### 2.3. Oral Indexes

Several oral indexes were studied. For the Plaque Index and the Gingival Index the evaluated teeth were usually numbered as “16”, “21”, “24”, “36”, “41”, “44”. Patients without any of the above-mentioned teeth were not included in the study. For other indexes, other present teeth were also used.

The Plaque Index (PI) estimates the area of a tooth covered by plaque by assigning a code: 0 = Tooth surface free from plaque; 1 = No plaque observed with the naked eye, but it can be extracted with a probe from the gingival crevice; 2 = Thin to a moderately thick layer of plaque covering the gingival area of the tooth; and 3 = Heavy layer of plaque covering the gingival area and the adjacent tooth [15].

The Gingival Index (GI), estimates the level of gingival tissue inflammation with a code: 0 = Absence of inflammation; 1 = Mild inflammation without bleeding while probing; 2 = Moderate inflammation with bleeding to probing; and 3 = Severe inflammation with spontaneous bleeding [15].

The Decayed, Missing, and Filled Teeth Index (DMF) is used to assess an individual’s oral health status based on frequency and intensity of cariogenic presence [16].

The Community Periodontal Index (CPI) assesses the periodontium status, conveying a code: 0 = “Healthy gum”; 1 = “Presence of bleeding on probing”; 2 = “Presence of supra or subgingival calculus”; 3 = “Periodontal pocket of 4 to 5 mm in depth”; and 4 = “Periodontal pocket 6 mm in depth” [17,18,19].

The Ligament Loss is most reliable when recorded immediately after CPI assessment. It identifies the severity of periodontal destruction classified by: 0 = “Insertion loss lesser than 1 mm” (healthy periodontal tissues); 1 = “Insertion loss between 1 and 2 mm” (mild periodontitis); 2 = “Insertion loss between 3 and 4 mm” (moderate periodontitis); and 3 = “Insertion loss greater than 5 mm” (severe periodontitis) [20].

The indices were calibrated by a team of specialist professors, from the Egas Moniz University Dental Clinic; this calibration was carried out by two professors, having observed the same patients. During this process, the professors didn’t have access to each other’s results. After a detailed analysis of the results obtained by each of the professors, a protocol was prepared. This protocol contained the mentioned codes, and the indices were always calculated by the same observer.

### 2.4. Data Collection

The clinical evaluation was carried out by two researchers using the same method, and recorded in the clinical files of the outpatient Clinic of Artificial Feeding. Before the study, both observers were extensively trained in the function using the same methodology and the same criteria. The observation periods of each researcher were sequential, but for each patient, T0 and T1 observations were performed by the same researcher. On the day of the endoscopic gastrostomy, just before the procedure (T0), data relating to the patients’ oral hygiene habits were recorded in a questionnaire adapted for a Portuguese population, from the Oral Health Questionnaire for Adults [21], which included identification, age, and gender, as well as the underlying disease that advocated for PEG. The participants answered all of the questions that were asked themselves and/or by their representatives, upon incapacity. At the T0 evaluation, a clinical observation was also conducted on the oral health parameters, namely, PI, GI, CPI, LL, and DMF. At the third month outpatient appointment (T1), data were again collected through clinical observation of PI, GI, CPI, LL, and DMF, to compare with the data obtained at T0. During each observation, the patient was seated in an armchair or lying on a gurney, in a room with both natural light and artificial light facing the mouth and using a periodontal probe to assess PI, GI, CPI, and LL; a basic kit consisting of a mirror, exploratory probe, and tweezers was also used to measure the DMF index. There were two data collection forms per patient, one for each observational moment (T0 and T1).

The clinical evaluation was carried out by two researchers using the same method.

### 2.5. Statistical Analysis

All statistical analyses were performed using the Statistical Package for Social Science (IBM^®^ SPSS^®^) version 26.0 software, with test results acknowledged as significant whenever the associated *p* < 0.05.

The changes observed in the oral cavity after three months of exclusive enteric feeding were evaluated by comparing the original oral index measurements (T0) with the ones taken at the follow-up (T1) through repeated measures generalized linear models. In these models, an independent dichotomous follow-up variable evaluated the changes over the study period while the dependent variable was assigned with each oral index. Therefore, the coefficient associated to the follow-up variable adjusted in the models reflected the effect size of the change, and the associated *p*-value the statistical significance of the change. Within this model framework, a linear model was adjusted for the continuous indexes, GI and PI, while a log-linear Poisson model was used for the indexes D, M, F, and DMF, as they result from counts. Furthermore, for the ordinal categorical indexes, AL and CPI, we adjusted an ordinal logistic model.

Regarding the influence of age, type of hygiene, pathology, and sex in the initial measures of the indexes, and in the contrasts between the baseline and follow-up measures, these variables were included as independent covariates in generalized linear models for fixed measures.

## 3. Results

### 3.1. Patients

The total initial sample included 40 patients, 20 (50%) female and 20 (50%) male. One patient died during the period between observations. Therefore, for analysis purposes, the final sample comprised 39 patients. Of these, 20 were female (51.3%), and 19 were male (48.7%). The age range was 23–90 years (mean: 65.3 ± 17.4, median: 69) and 25 patients were over 65 years old.

About three quarters (*n* = 29, 74.4%) presented dysphagia due to neurological disorders. The remaining 10 (25.64%) suffered from head or neck cancer. Stroke was the most frequent disease (*n* = 11, 28.21%), followed by head or neck cancer (*n* = 10, 25.6%) and dementia (*n* = 9, 23.1%). Three quarters (*n* = 30, 76.9%) were dependent on caregivers for oral hygiene (Table 1).

### 3.2. Oral Health Questionnaire

Only 36 out of 39 patients answered the questionnaire. Neurological disorders made it impossible for three patients to answer the questions. The majority of our sample reported brushing their teeth with fluoride toothpaste once a day to twice or more times a day. Despite mouth rinsing not being a usual practice among the patients, they used gauze soaked in mouthwash alone or in addition to the toothbrush. In contrast to the self-reporting of good oral hygiene, most of the participants reported dissatisfaction concerning the state of their oral health and that they had not attended a dentist appointment during the last year (Table 2).

### 3.3. Plaque Index

The mean value of PI in T0 was 1.840.59, while it was 1.870.65 in T1. Despite this, the clinical decline had a *p*-value of 0.836 and did not reach a statistically significant difference between the two monitoring periods (Table 3).

### 3.4. Gingival Index

The GI mean value was 1.590.70 at the T0 examination and 1.650.67 at the T1 evaluation. The clinical deterioration was not statistically significant (Table 4).

### 3.5. Decayed, Missing, and Filled Teeth Index

At the T0, the DMF teeth index presented a mean of 16.08 ± 7.21 and in T1, it was registered as 16.36 ± 7.19.

In the T0, we can account for 175 (28%) decayed teeth (Mean 4.49 ± 3.75), 421 (67%) missing teeth (Mean: 10.79 ± 6.69), and 31 (5%) filled teeth (Mean: 0.79 ± 7.21). At T1, there were 186 (29%) decayed teeth (Mean: 4.77 ± 3.90), 424 (66%) missing teeth (Mean: 10.87 ± 6.80), and 28 (4%) filled teeth (Mean: 0.72 ± 1.49).

Concerning the overall DMF teeth index, there was a significant statistical difference between both observational times (*p* = 0.007). Regarding each parameter, the significance was registered for the number of Decayed Teeth (*p* = 0.042) (Table 5).

It was found that none of the parameters, gender, type of pathology, hygiene, and age, had a relevant statistical effect on the DMF index (Table 6).

### 3.6. Periodontal Index and Ligament Loss

According to CPI, at T0, only one patient presented gingival health (2.6%), seven had bleeding on probing (18%) and four had supra or subgingival calculus (10%). Seventeen patients had a pocket with 4 to 5 mm in depth (44%) and eight with 6 mm or more (21%). It was not possible to classify two individuals (5%) according to the score. At the T1 evaluation, no individual showed gingival health, seven had bleeding upon probing (18%), and five had either supra or subgingival calculus (13%). Sixteen presented a pocket of 4 to 5 mm (41%) and nine with 6 mm or more (23%). It was also impossible to determine the score of two patients (5%) due to insufficient collaboration.

Regarding LL at the T0, eight patients presented healthy periodontal tissues (20.5%), seven presented mild periodontitis (18%), sixteen demonstrated moderate periodontitis (41%), and eight displayed severe periodontitis (21%). At the T1 evaluation, eight patients presented healthy periodontal tissues and sixteen presented moderate periodontitis. Six presented mild periodontitis (15.4%) and nine presented severe periodontitis (23%).

For either index, there was no significant statistical difference found, between T0 and T1 (Table 7).

## 4. Discussion

### 4.1. Patients

Oral health of PEG-patients is an important issue still poorly addressed. Longitudinal studies with PEG-patients are scarce due to challenges in long term follow-up. These patients suffer from devastating diseases and a large number survive only a few months after the gastrostomy procedure. Those who live longer are typically reliant on family caregivers or nursing homes, transportation to an outpatient clinic can be challenging, and a large number are lost for follow-up.

These challenges were critical while selecting patients for the study. Social conditions should favor compliance with the follow-up and the clinical settings should be stable enough to assure a very probable three months survival. In fact, a three-month interval between each observation was chosen because it matches with a moment of the standard follow-up (standard outpatient appointments are at one, three, and six months after the endoscopic procedure). At one month it would be unlikely to observe important changes in oral features. Many patients skip their six-month appointments for a variety of reasons, including death, relocation to a nursing facility distance from the hospital, or simply being stable and ignoring the appointment.

Three months was thought to be a good compromise for the current study’s follow-up evaluation.

The initial convenience sample was composed of 40 individuals, however, one patient died between the first and the second observation moments. PEG is a relatively safe and predictable procedure [22], but is associated with a high mortality rate, which is mostly due to the underlying disorders and not the procedure itself [23]. This is foreseeable, as the most of PEG-patients from the HGO’s Artificial Nutrition Consultation are senior citizens [24] and suffer from age-related underlying pathologies [25]. Furthermore, dysphagia caused by neurological disorders is the most common indication for PEG [26,27], similar to our sample. Globally, our group of patients was comparable to the patients followed by all teams taking care of gastrostomy fed patients, either looking at ages, sex, or underlying diseases.

Most of these patients were dependent on caregivers as they had difficulty in performing oral hygiene themselves due to loss of cognitive ability and motor functions, as described in similar clinical settings [28]. Gastrostomy patients are often exposed to a greater risk of dental caries as caregivers usually have little knowledge concerning oral health [29]. This may be caused by demotivation or personal psychological barriers [30] and the increased workload that caregivers endure during stressful situations [31]. For some head and neck cancer patients, radiotherapy may have been another cariogenic factor. Therefore, less appropriate hygiene is provided to the patients [32], even when oral health care is deemed important by the caregivers [31].

### 4.2. T0 Clinical Questionnaire

Missing teeth was prevalent in most of the patients included in this study, which may be correlated to aging as it was mentioned by Fleming who described tooth loss to be predominant in adults over 65 years old, and that it increased with age [33] due to inadequate control of chronic diseases, poor oral hygiene, and low attendance to oral health care services [34]. As a result, difficulty in chewing, swallowing, and communication can occur [35] and, if not treated, it may evolve to a psychological and social level [36], negatively affecting quality of life [37].

Over half of this sample consider their teeth health to be “very poor”/“poor”, but a minority perceives them as “good”/”very good” and showed similar appreciation for their gums. Self-perception of oral health does not reflect the true clinical status and some answers indicated the misperception of patients and caregivers, keeping with a previous study, which demonstrated poorer self-reported oral health and expressed lack of awareness from both. Patients may have stated a positive opinion even if their oral health was impoverished [38]. Not perceiving the real degradation of oral health is another important factor for the persistently insufficient dental care.

The majority of patients had a hygiene practice that was clearly below the desired standard. Half of the subjects reported cleaning their teeth only once a day and others at least twice, while a minority reported it once a week or never at all. A toothbrush was the most used hygiene aid followed by a gauze soaked in mouthwash. Mouth rinsing is not a safe cleaning method for soft tissues in dysphagic patients; nevertheless, chlorhexidine could be applied in a gel form or with a sponge [39], but, in fact, it was not generally used. Our patients used fluoride toothpaste, which is indicated for tube fed patients, preferably with no foam-inducing agents to reduce the risk of aspiration [39].

A small portion of these patients wore dentures with the intent of regaining chewing and leading to restoring cognitive function, self-esteem, and quality of life [40]. In our study, a normal toothbrush was the most common item used to brush dentures, often used once a day or never at all. Toothpaste was deemed the most frequent cleaning agent followed by just water, similar to what was described in a previous article concerning dental health professional recommendations and consumer habits in denture cleansing. In that study, cleanser tablets were the most recommended cleaning agent and freshwater was mostly recommended for rinsing and soaking [41].

Regarding oral hygiene habits, most patients changed their routines as a consequence of the disorders. Neurological disorders are recognized to reduce the quality and management of self-care, contributing to biofilm accumulation, the emergence of periodontal disease [42,43], and increased risk of caries [44]. Lyra’s study population also reflected a high prevalence of periodontitis and gum inflammation, due to the fine motor issues and cognitions deficits expected from the disorder [43]. This can become another cariogenic factor in neurological PEG patients.

Just a quarter of our patients reported having attended a dental appointment during the last 6 months before gastrostomy. Most of which went to those appointments sometime between 2 to 5 years. The major reason given for the consultation was “treatment or follow-up treatment”, then “pain or problems with their teeth, gums or mouth”. Nevertheless, these patients that attended dental appointments did not present better oral health, as described in Carvalho’s study. To her understanding, half of the population group did not visit the dentist in over a year and those who did tended to be more sensitive to the reality of their oral health status [45]. Long before starting on enteral feeding, our patients tend to visit the dentist only when they have pain or discomfort, further aggravating their oral health. In fact, poor oral health was present before the gastrostomy and PEG feeding worsened the process.

### 4.3. Oral Indexes: T0

At the beginning of the study, the PI index indicated moderate dental plaque on the gingival area surrounding teeth. The GI index also presented moderate values corresponding to mild inflammation with no bleeding upon probing. Oral hygiene shifts the oral microbiota and, subsequently, is in control of the patient’s inflammatory response [46]. Hence, these two indexes are correlated and should present similar results.

Our sample presented an initial mean for the DMF index equal to 16.08, higher than the mean presented by the Portuguese population between 60 and 75 years old (15.11) [47]. Therefore, from the beginning of our study patients presented worse dental status than the national average.

Concerning the initial periodontal status, 25 of our patients (65%) suffered from moderate to severe periodontitis. Conversely, the majority of the Portuguese population aged 60 to 75 years old do not present periodontal disease [47].

Altogether, T0 Oral Indexes of our sample display a poor dental health, worse than that observed in the Portuguese population of the same age, reflecting the previous widespread lack of self-care.

### 4.4. Oral Indexes: T0–T1 Evolution

Upon the second observational moment, a small increase in PI was noticeable. Previous studies have demonstrated that PEG-patients show more accumulation of plaque than non-PEG groups [48]. The absence of chewing could result in a decrease in saliva production, otherwise responsible for cleaning the mouth, allowing for plaque deposition [49], which is further exacerbated by the frequent neglect of oral hygiene in these patients [50]. Subsequently, this may increase the risk of bronchial aspiration and even bronchopneumonia [48].

Clinically, a slight increase in GI values was observed, showing mild inflammation in accordance with a study showing that tube-fed children present mild to moderate gingivitis [51]. Furthermore, PEG patients present similar gingival inflammation to those not undergoing gastrostomy [52]. Thus, our patient’s inflammatory state may result from the irritation caused by the deposition of bacterial plaque linked to their poor oral hygiene, and this low-grade inflammatory state is a contribution to global health impairment.

Our sample’s DMF index presented a higher value at the second observation moment, mostly because of the growing number of decayed teeth. Feeding tubes should advocate for a lower caries index as indicated in Nasu’s study, since it does not allow for oral intake; nonetheless, a major reason for this observation should be a good oral hygiene practice [51], because the absence of chewing is responsible for a reduction in saliva [53], increasing the risk of decay, erosion, and xerostomia [54]. Furthermore, hyposalivation can also be linked to defective salivary gland function due to neurological disorders [55], present in our sample.

As it was verified by T1, there was a low opportunity of aggravating our patient’s periodontal condition since they already presented worse periodontal status compared to the national Portuguese average. Patient’s Periodontal Disease may be promoted by the constant deposition of plaque in the subgingival space described with PEG-patients [48], and also by their tobacco and alcohol consumption, which is related to poor oral health.

Overall, between the two observational moments, all indexes deteriorated. Although just some of the registered deterioration of these oral health indexes reached statistical significance, the clinical evaluation of the present results is very clear; due to lack of chewing and swallowing ability, combined with increased limitations to oral and dental care, PEG-patients suffer an overall deterioration of oral health, worsening a previously concerning oral health, impaired by former poor hygiene habits. Professional oral health support for these patients should have a special focus on those persons that are expected to resume oral feeding and on those who may become PEG-fed long survivors. Dentist and/or dental hygienist appointments should become part of standard care during PEG-patients follow-up, in order to improve oral health and general health.

### 4.5. Study Limitations

The present study used a convenience sample from a single center, although it was the larger artificial feeding center in Portugal. A larger sample was not possible due to several issues. Many patients in the Artificial Nutrition Clinic were edentulous, others had difficulties in opening their mouths, and others had behavioral changes that limited them from participating safely. We had also taken into account the poor attendance of some patients in the Artificial Nutrition Outpatient Clinic. Furthermore, this is a group with a high mortality rate. Examination not in the dentist’s office, without the possibility of drying the teeth when assessing indicators (especially DMF), is also a limitation.

Nevertheless, the present study obtained important clinical data, regarding the problem of oral health of PEG-patients.

## 5. Conclusions

The present study highlights the weakened oral health of PEG-patients. When compared to the Portuguese population, these patients present with impaired oral health indexes, as this is present even before the gastrostomy procedure associated to reduced general health. During the three months of exclusive enteral feeding, without any oral ingestion, all studied oral features worsened, likely due to the absence of oral physiological activity, the declining general health, and the reduced or absent oral hygiene. A better focus on oral hygiene of PEG-patients is recommended, especially for those with a longer life expectancy and regular dentist and/or dental hygienist appointments, as part of standard follow-up.

## Figures and Tables

**Table 1 ijerph-19-04827-t001:** Socio-demographic characteristics of the participating subjects from the study.

Variables	Min	Max	Mean	Median	SD
Age	23	90	65.3	69	17.4
Sex	Female 20 (51.3%)		Male 19 (48.7%)		
Underlying Disorder	Neurological Disorders 29 (74.4%)		Head or Neck Cancer 10 (25.6%)		
	Dementia 9 (23.1%)				
	Stroke 11 (28.2%)				
	Amyotrophic Lateral Sclerosis 3 (7.7%)				
	Anoxic Encephalopathy 1 (2.6%)				
	Subarachnoid Hemorrhage 1 (2.6%)				
	Cerebral Palsy 2 (5.1%)				
	Parkinson 1 (2.6%)				
	Brain Trauma 1 (2.6%)				
Oral Hygiene dependence	Non-dependent 9 (23.1%)		Dependent 30 (76.9%)		

**Table 2 ijerph-19-04827-t002:** Oral health questionnaire.

Questions	Answers							
How many natural teeth do you have?	No natural teeth 0 (0%)		1–9 teeth 5 (13.9%)		10–19 teeth 15 (41.7%)	20 teeth or more 16 (44.4%)		
During the past 12 months, did your teeth or mouth cause any pain or discomfort?	No 16 (44.4%)		Yes 14 (38.9%)		Don’t know 5 (13.9%)	No answer 1 (2.8%)		
Do you have any removable dentures?	No 28 (77.8%)	Partial upper denture 5 (13.9%)	Partial lower denture 0 (0%)		Full upper denture 3 (8.3%)	Full lower denture 0 (0%)		
How would you describe the state of your teeth?	Excellent 0 (0%)	Very good 1 (2.8%)	Good 4 (11.1%)	Average 7 (19.4%)	Poor 10 (27.8%)	Very Poor 13 (36.1%)		Don’t know 1 (2.8%)
How would you describe the state of your gums?	Excellent 0 (0%)	Very good 2 (5.6%)	Good 5 (13.9%)	Average 7 (19.4%)	Poor 9 (25%)	Very Poor 12 (33.3%)		Don’t know 1 (2.8%)
How often do you clean your teeth?	Never 2 (5.6%)	Once a month 0 (0%)	2–3 times a month 0 (0%)	Once a week 1 (2.8%)	2–6 times a week 0 (0%)	Once a day 18 (50%)		Twice or more a day 15 (41.7%)
Do you use any of the following items to clean your teeth?	Toothbrush 31 (91.2%)	Wooden toothpicks 0 (0%)	Plastic toothpicks 0 (0%)	Dental floss 0 (0%)	Charcoal 0 (0%)	Interdental brush 0 (0%)	Chewstick/miswak 0 (0%)	Gauze soaked mouthwash 11 (32.4%)
Do you use toothpaste to clean your teeth?	Yes 27 (79.4%)				No 7 (20.6%)			
Do you use toothpaste that contains fluoride?	Yes 19 (70.4%)			No 0 (0%)		Don’t know 8 (29.6%)		
Do you use mouthwash?	Yes 7 (19.4%)				No 29 (80.6%)			
How often do you brush your dentures?	Never 3 (37.5%)		Once a day 3 (37.5%)		Twice a day 1 (12.5%)		3 or more times a day 1 (12.5%)	
Do you use any of the following items to brush your dentures?	Denture brush 0 (0%)		Toothbrush 5 (62.5%)		Other 0 (0%)		I do not brush 3 (37.5%)	
What do you use to clean your dentures?	Water 1 (12.5%)	Soap 0 (0%)	Denture cleanser 0 (0%)		Toothpaste 5 (62.5%)		No answer 2 (25%)	
How long is it since you last saw a dentist?	Less than 6 months 3 (8.3%)	6–12 months 6 (16.7%)	More than 1 year but less than 2 years 5 (13.9%)		2 years or more but less than 5 years 14 (38.9%)	5 years or more 8 (22.2%)	Never received dental care 0 (0%)	
What was the reason for your last visit to the dentist?	Consultation/advice 2 (5.6%)	Pain or trouble with teeth, gums or mouth 8 (22.2%)		Treatment/follow-up treatment 19 (52.8%)		Routine check-up/treatment 6 (16.7%)		Don’t know/don’t remember 1 (2.8%)
Have you maintained the same oral hygiene habits throughout your life?	Yes 14 (38.9%)				No 22 (61.1%)			

**Table 3 ijerph-19-04827-t003:** Descriptive and Comparative analysis of Plaque Index, between observational assessments.

Plaque Index ^a^	Min	Max	MV ± SD	Sig.
T0	0.13	3	1.84 ± 0.68	F = 0.044; *p* = 0.836 *
T1	0.10	3	1.87 ± 0.68	

^a^ Linear model for continuous indexes. Min: Minimum value; Max: Maximum value; MV: Mean value; SD: Standardized Deviation; Sig.: Statistical significance. * F test for marginal means.

**Table 4 ijerph-19-04827-t004:** Descriptive and comparative analysis of Gingival Index, between observational assessments.

Gingival Index ^a^	Min	Max	MV ± SD	Sig.
T0	0	3	1.59 ± 0.70	F = 0.433; *p* = 0.514 *
T1	0.10	3	1.65 ± 0.67	

^a^ Linear model for continuous indexes. Min: Minimum value; Max: Maximum value; MV: Mean value; SD: Standardized Deviation; Sig.: Statistical significance. * Test F for marginal means.

**Table 5 ijerph-19-04827-t005:** Descriptive and comparative analysis of the Decayed, Missing, and Filled Teeth Index and its individual parameters, between assessment moments.

DMF ^b^	Min	Max	MV ± SD	Sig.
T0	1	28	16.08 ± 7.21	Χ^2^ = 7.611; *p* = 0.007 *
T1	1	28	16.36 ± 7.19	
Decayed ^b^	*n*	%		
T0	175	28	4.49 ± 3.75	Χ^2^ = 4047; *p* = 0042 *
T1	186	29	4.77 ± 3.90	
Missing ^b^				
T0	421	67	10.79 ± 6.69	Χ^2^ = 1.026; *p* = 0.301 *
T1	424	66	10.87 ± 6.80	
Filled ^b^				
T0	31	5	0.79 ± 7.21	Χ^2^ = 3.250; *p* = 0.071 *
T1	28	4	0.72 ± 1.49	

^b^ Poisson’s log-linear model for data resulting from counts. DMF: Decayed, Missing, and Filled teeth index; Min: Minimum value; Max: Maximum value; MV: Mean value; SD: Standardized Deviation; Sig.: Statistical significance; *n*: Frequency; %: Percentage. * Wald Qui-square.

**Table 6 ijerph-19-04827-t006:** Inferential analysis on the influence of sex, type of pathology, hygiene, and age in the Decayed, Missing, and Filled teeth index variations, between assessment moments.

DMF	Sig.
Sex	Χ^2^ = 2.324; *p* = 0.127 *
Age	Χ^2^ = 0.096; *p* = 0.756 *
Oral Hygiene	Χ^2^ = 0.097; *p* = 0.756 *
Type of Pathology	Χ^2^ = 0.015; *p* = 0.903 *

DMF: Decayed, Missing, and Filled teeth index; Sig: Statistical Significance. * Wald’s Qui-square.

**Table 7 ijerph-19-04827-t007:** Descriptive and comparative analysis of Periodontal Index and Ligament Loss, between assessment moments.

**Community Periodontal Index ^c^**	**T0**	**T1**	**Sig.**
Gingival health	1 (2.6%)	0 (0%)	Χ^2^ = 0.960; *p* = 0.327 *
Bleeding upon probing	7 (18%)	7 (18%)	
Presence of supra or subgingival calculous	4 (10%)	5 (13%)	
Periodontal pocket with a depth of 4 to 5 mm	17 (44%)	16 (41%)	
Periodontal pocket with a depth equal to or greater than 6 mm	8 (21%)	9 (23%)	
Unable to determine	2 (5%)	2 (5%)	
**Ligament Loss ^c^**	**T0**	**T1**	**Sig.**
Attachment loss less than 1 mm (Healthy Periodont)	8 (20.5%)	8 (20.5%)	Χ^2^ = 2.033; *p* = 0154 *
Attachment loss between 1 and 2 mm (Mild Periodontitis)	7 (18%)	6 (15.4%)	
Attachment loss between 3 and 4 mm (Moderate periodontitis)	16 (41%)	16 (41%)	
Attachment loss greater than 5 mm (Severe Periodontitis)	8 (21%)	9 (23%)	

^c^ Logistic ordinal model for ordinal categorical data. Min: Minimum value; Max: Maximum value; MV: Mean value; SD: Standardized Deviation; Sig.: Statistical significance. * Wald Qui-square.

## Data Availability

The data presented in this study are available on request from the last author.

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
