# Peer review of "Oral Health Status of Adult Dysphagic Patients That Undergo Endoscopic Gastrostomy for Long Term Enteral Feeding"

_ijerph, 2022, doi:10.3390/ijerph19084827_

Round 1
Reviewer 1 Report
The authors responded to the comments in the review and significantly improved the manuscript.
Author Response
Dear Reviewer,
Thank you for your feedback
Reviewer 2 Report
The authors presented a cohomprensive prospective study that was correctly run out and revised。
Author Response

(The authors gave the same response as above.)

Reviewer 3 Report
The authors aimed to evaluate the oral health and oral changes of Percutaneous endoscopic gastrostomy (PEG)-patients without oral feeding for three months.
This study tried to contribute to the understanding of oral health in PPEG-subjects, to raise awareness of the importance of this issue, and to identify strategies to minimize the degradation of oral health in patients that undergo PEG-feeding.
The study covers some issues that have been overlooked in other similar topics. The structure of the manuscript appears adequate and well divided in the sections. Moreover, the study is easy to follow, but few issues should be improved. Some of the comments that would improve the overall quality of the study are:
- Authors must pay attention to the technical terms acronyms they used in the text.
- English language needs to be revised.
- Conclusion Section: This paragraph required a general revision to eliminate redundant sentences and to add some "take-home message".
Author Response
Dear Reviewer
Thank you for your kind words and useful comments.
In order to improve our manuscript the authors followed your suggestions:
- Authors must pay attention to the technical terms acronyms they used in the text.
All the technical terms acronyms were reviewed by a dentistry associated professor.
- English language needs to be revised.
The manuscript was reviewed by an English native speaker from Canada, that performed a large and diffuse correction of the paper’s language.
- Conclusion Section: This paragraph required a general revision to eliminate redundant sentences and to add some "take-home message".
The Conclusion Section underwent a general revision in order to became more concise and allowing the Conclusion to be read like a "take-home message", of the present study.
All changes are highlighted in the manuscript.
The authors bilieve that, following the Reviewer’s suggestions, the actual manuscript is now more interesting for IJERPH readers.
Thank you so much,
The authors

This manuscript is a resubmission of an earlier submission. The following is a list of the peer review reports and author responses from that submission.
Round 1
Reviewer 1 Report
Oral health in patients with systemic diseases is an important issue. The authors highlighted an oral health problem that is often overlooked when treating the underlying disease in these patients.
Abstract
Line 17 : This study to aimed evaluate instead This study aimed evaluate
Line 21: abbreviation- Decayed, Missing and Filled Teeth Index (DMF)
Introduction
The influence of oral health on general health in patients with systemic diseases? What can be the negative effects of oral problems on the general health in these patients?
Materials and Methods
The authors should describe the method of examining the indicators and their calculation. Which teeth should be examined according to the indices and how they assessed in patients with missing teeth?
Line 111: CPI assesses the periodontium status , the needs are determined by the index of treatment needs (TN).
What was the specialization of the person who examined the patients ? Dental specialist?
Results
39 patients participated in the study. Why only 36 people participated in the survey?
Tab.7 -CPI -why Unable to determine in 2 patients?
Attachment loss less than 1mm instead Attachment loss lesser than 1mm
Discussion
The authors mentioned that there are few publications on this subject, but the discussion should refer to these works. What results have other authors obtained? Compare to your own results.
The authors might consider adding a sentence about the importance of including oral health professionals in multi-disciplinary teams involved in the care of these patients.
Study limitations
Examination not in the dentist's office, without the possibility of drying the teeth when assessing indicators (especially DMF), is also a limitation.
References
self-citations
8-12,14
Author Response
Dear Sir,
Thank you very much for your revision and important suggestions. We appreciate the reviewer comments and will respond point by point immediately below.
Oral health in patients with systemic diseases is an important issue. The authors highlighted an oral health problem that is often overlooked when treating the underlying disease in these patients.
Thank you very much for your comment.
Abstract
Line 17 : This study to aimed evaluate instead This study aimed evaluate
This was correct.
Line 21: abbreviation- Decayed, Missing and Filled Teeth Index (DMF)
This was correct.
Introduction
The influence of oral health on general health in patients with systemic diseases? What can be the negative effects of oral problems on the general health in these patients?
Poor oral health results on low grade inflammation which is associated with worst outcomes in a wide range of systemic disorders. This has been included in the text.
Materials and Methods
The authors should describe the method of examining the indicators and their calculation. Which teeth should be examined according to the indices and how they assessed in patients with missing teeth?
For the Plaque Index and the Gingival Index The evaluated teeth were the usually numbered as 16 21 24 36 41 44. No patient was included missing any of those teeth. For other indexes other present teeth were also used. This was included in the manuscript.
Line 111: CPI assesses the periodontium status , the needs are determined by the index of treatment needs (TN).
This was correct.
What was the specialization of the person who examined the patients ? Dental specialist?
The examinators were two last year dentistry master students supervised by a dentistry professor.
Results
39 patients participated in the study. Why only 36 people participated in the survey?
Only 36 out of 39 patients answered the questionnaire. Neurologic disorders prevent 3 patients from answering. This was added to the text.
Tab.7 -CPI -why Unable to determine in 2 patients?
It was impossible to determine 2 patients due to insufficient collaboration. This was added to the text.
Attachment loss less than 1mm instead Attachment loss lesser than 1mm
This was correct.
Discussion
The authors mentioned that there are few publications on this subject, but the discussion should refer to these works. What results have other authors obtained? Compare to your own results.
The authors might consider adding a sentence about the importance of including oral health professionals in multi-disciplinary teams involved in the care of these patients.
Discussion now states: Dentist and/or dental hygienist appointments should become part of standard care during PEG-patients follow-up, in order to improve oral health and general health.
Study limitations
Examination not in the dentist's office, without the possibility of drying the teeth when assessing indicators (especially DMF), is also a limitation.
This is an important limitation and was included. Thank you so much.
References
self-citations
8-12,14
It is certainly research from our team. But, in fact, in the last decade our team produced an important bulk of research on several under evaluated issues of PEG patients and the present study provides a research issue in continuity with the perspective of improving global health of PEG patients.
Globally, your suggestions greatly improved our manuscript.
Thank you very much.

Reviewer 2 Report
Dear author
A good attempt to collect data, however, it requires a lot of correction. missing patient data and grammatical errors please improve upon the required aspects.

Author Response
Dear Sir,
Thank you very much for the careful review and the important suggestions. We believe that we have responded to most of these suggestions (in the manuscript) and we believe that, now, the manuscript is very enriched and become much more interesting and more useful for our readers.
Thank you very much

Round 2
Reviewer 1 Report
Thank you for your explanation and changes.
Authors should improve the discussion and compare their results to those of other authors cited in references.

Author Response
Reviewer 1 (round2)
Comment: Authors should improve the discussion and compare their results to those of other authors cited in references.
We want to thank the reviewers for their comments and efforts towards improving our manuscript. We have incorporated changes to reflect your suggestions. We have highlighted the changes within the manuscript.

Reviewer 2 Report
Dear Authors
The changes were recommended satisfactorily done. The article looks more readable and authentic.
Author Response
We want to thank the reviewer for their comments and efforts towards improving our manuscript